# Incidental Pathologic Findings from Orthodontic Pretreatment Panoramic Radiographs

**DOI:** 10.3390/ijerph20043479

**Published:** 2023-02-16

**Authors:** Phumzile Hlongwa, Mpule Annah Lerato Moshaoa, Charity Musemwa, Razia Abdool Gafaar Khammissa

**Affiliations:** 1Department of Orthodontics, School of Dentistry, Faculty of Health Sciences, University of Pretoria, Pretoria 0001, South Africa; 2Department of Orthodontics, School of Oral Health Sciences, Faculty of Health Sciences, University of the Witwatersrand, Johannesburg 2050, South Africa; 3Department of Periodontology and Oral Medicine, School of Dentistry, Faculty of Health Science, University of Pretoria, Pretoria 0001, South Africa

**Keywords:** incidental pathologic findings, orthodontic pre-treatment, panoramic radiograph

## Abstract

Panoramic radiography is frequently performed for new patients, follow-ups and treatment in progress. This enables dental clinicians to detect pathology, view important structures, and assess developing teeth. The objective of the study was to determine prevalence of incidental pathologic findings (IPFs) from orthodontic pretreatment panoramic radiographs at a university dental hospital. A retrospective cross-sectional review was conducted of pretreatment panoramic radiographs, using data collection sheets with predefined criteria. Demographic data and abnormalities (impacted teeth, widening of periodontal ligament, pulp stones, rotated teeth, missing teeth, unerupted teeth, crowding, spacing, supernumerary teeth, and retained deciduous teeth) were reviewed. SPSS 28.0 was used to analyze data with statistical tests set at a 5% significance level. Results: One hundred panoramic radiographs were analyzed with an age range of 7 to 57 years. The prevalence of IPFs was 38%. A total of 47 IPFs were detected with altered tooth morphology predominantly (n = 17). Most IPFs occurred in males (55.3%), with 44.7% in females. A total of 49.2% were in the maxilla and 50.8% in the mandible. This difference was statistically significant (*p* < 0.0475). Other abnormalities were detected in 76% of panoramic radiographs; 33 with IPFs and 43 without. A total of 134 other abnormalities detected showed predominantly impacted teeth (n = 49). Most of these abnormalities were in females (n = 77). Conclusions: The prevalence of IPFs was 38%, predominated by altered tooth morphology, idiopathic osteosclerosis, and periapical inflammatory lesions. Detection of IPFs from panoramic radiographs underscored the importance for clinicians to examine them for comprehensive diagnosis and treatment planning, especially in orthodontics.

## 1. Background

Panoramic radiography is a radiologic technique that produces an image of the facial structures, including the mandibular and maxillary dental arches and their supporting structures [1,2]. A panoramic radiograph is a two-dimensional (2D) radiograph that can detect pathologies or abnormalities in structures of the maxillofacial complex [3]. Panoramic radiographs have a low radiation dose for the patient [1,4,5], can be used in patients who are not able to open their mouths [6], are convenient when examining a patient, and are quick to take [7].

In orthodontics, panoramic radiographs are taken routinely to detect malocclusions [8] and to assess mesiodistal root angulations, before, during, and after orthodontic treatment as a guide in establishing proper root position [1,5]. They also provide information about the tissues surrounding the teeth, axial inclinations, and maturation periods [9].

Over the years, questions have been raised about the role and level of responsibility of orthodontists in discovering incidental pathologies [10]. Orthodontists should be mindful of the possibility of the existence of pathology in orthodontic patients and should be able to identify such pathologies on radiographs [11]. Kuhlberg and Norton have underscored the importance of assessing orthodontic pretreatment radiographs for pathologies [8].

Any pathologic or abnormal findings that emerge, independent of the original reason for taking radiographs, are defined as incidental pathology findings (IPFs) [3]. To prevent patient mismanagement, correct diagnosis of IPFs or odd normal variants is essential and requires knowledge about the anatomy and pathology of the head and neck area [12]. The chances that dentists will be able to discover incidental findings of anomalies or pathologies from panoramic radiographs of patients will be high if they are calibrated and also have a special interest in the subject [13]. It has been suggested that there is a chance that orthodontists might miss incidental findings owing to being distracted by areas of interest in a radiograph [14].

Jose and Varghese recommend that panoramic radiographs must be taken before any dental procedure to detect occult pathologies which have nothing to do with the presenting complaint. There is a 43% chance that important dental anomalies are missed in the premaxillary region in a panoramic radiograph [15].

A study by Bondemark et al., reported a predefined criteria that can be used to assess IPFs, shown in Table 1 [13]. Their findings have emphasized the importance of analyzing panoramic radiographs not just from an orthodontic perspective but also for incidental pathological findings.

The level at which an orthodontist can identify incidental pathologic findings from an orthodontic patient is of importance because these findings might need odontological or medical intervention and management. Therefore, the aim of the study was to determine the incidental pathologic findings from orthodontic pretreatment panoramic radiographs.

## 2. Methods

A retrospective cross-sectional study was conducted using radiographs from 1 January 2015 to 31 December 2019. The study setting was the postgraduate clinic of the Department of Orthodontics at a university dental hospital. Utilizing all pretreatment panoramic radiographs as the sampling frame, systematic sampling was used to select a minimum of 100 radiographs. All panoramic radiographs were taken following a standard protocol with Orthophos XG3D/Ceph (Sirona Dental Systems, Germany) at an adjusted voltage of 60–90 Kv, and 3–16 mA, and an exposure time of 9.4–14.1 s. The Sirona protocol was adjusted to the patient’s age, size, and weight.

Sample size was calculated using a prevalence of 8.7% in incidental pathological findings in the orthodontic pre-treatment panoramic radiographs reported by Bondemark et al. [13]. A 95% confidence level and a precision of 5% in the prevalence estimate using the following Dobson’s formula [16] to give a minimum sample size of:$$n=\frac{Z^{2}\times p(1-p)}{d^{2}}$$
where p is the prevalence of incidental pathological findings (8.7%);

d is the precision of the estimate of prevalence of incidental findings (0.05);

Z is the standard normal deviate at a given confidence level (1.96 at 95% confidence level).

Thus, using these input parameters, the minimum sample size was calculated to be
n = 1.96^2^ × (0.07) (1 − 0.07)/0.05^2^

n = 100

The inclusion criteria were patient clinical records with no previous history of orthodontic treatment. The age group included adults and children and panoramic radiographs of good quality were used. Patients with previous history of orthodontic treatment and craniofacial anomalies were excluded from the study. Ethical approval was obtained from the Human Research Ethics Committee (M210625) to conduct the study.

### 2.1. Data Collection

The type, location, and distribution of incidental pathological findings (IPFs) from the orthodontic pretreatment panoramic radiographs were recorded using the predefined criteria described by Bondemark et al. [13] shown in Table 1. The data was captured on a data collection sheet. Findings such as caries, missing/supernumerary teeth, impacted teeth, crowding, and eruption disturbances were not recorded as IPFs because they are generally noted in the dental assessment. However, they were recorded as other abnormalities that were seen in these radiographs.

A computer (Intel(R) Core™i5-7500 CPU @3.40 GHz 3.41 GHz) with a screen of 17.32 inches (Horizontal) and 9.84 inches (vertical) and 1600 × 900 (60 PHz) pixels was used to view the panoramic radiographs. Using the same lighting and environmental conditions, with no interruptions during a series of no more than 15 observations per session (2 h) to avoid visual fatigue of the operator, the radiographs were viewed from the SIRONA SIDEX XG™, a Sirona Dental X-ray and Imaging System software, to detect any IPFs.

Intra-observer and inter-observer reliability was assessed using the following Kappa statistic with its 95% confidence bounds:$$k=\frac{P_{0}-P_{c}}{1-P_{c}}$$

With 95% confidence interval: $k\;\pm \;1.96se\left(k\right)$

where,

$P_{0}$ is the proportion observed agreements

$P_{c}$ is the proportion agreements expected by chance

Intra-examiner reliability tests showed almost perfect agreement with Kappa scores greater than 0.85 for all types of findings analyzed. Estimates for inter-examiner reliability tests also showed a strong agreement with Kappa score greater than 0.81 for all types of findings analyzed.

### 2.2. Statistical Analysis

Statistical Package for the Social Sciences (SPSS Version 28.0 for Windows) was used for data analysis. The prevalence of IPFs was calculated; median and interquartile range (IQR) were used for age. Tests for normality were conducted using the Shapiro–Wilk test. To determine the relationship of IPFs with gender and location, the chi-square test was used. The Mann–Whitney U test was used for age. The significance level alpha (α) was set at 0.05 for all statistical tests.

## 3. Results

A total of 100 panoramic radiographs were analyzed, and IPFs were found in 38 (38%) of the radiographs. Table 2 shows the demographics of the sample indicating 41 males and 59 females. Forty-seven findings were detected from the 38 panoramic radiographs, 20 from males and 18 from females.

Table 2 shows a total of 47 findings that were detected from the 38 panoramic radiographs. Altered tooth morphology was the most predominant finding (n = 17), followed by idiopathic osteosclerosis (n = 13), then periapical inflammatory lesions (n = 8). Of the 47 findings that were detected, n = 26 (55.3%) were observed in males and n = 21 (44.7%) in females. Altered tooth morphology was predominant in both males and females, with idiopathic osteosclerosis more predominant in males. The findings were not statistically significant (*p* < 0.107).

Of the 38 panoramic radiographs that had IPFs, 30 had only one IPF ranging from periapical inflammatory lesion, marginal bone loss, cyst in the alveolar bone (Figure 1), idiopathic osteosclerosis, or thickening of the mucosal lining of sinuses. Altered tooth morphology was recorded as IPF in the absence of periapical or periodontal pathoses, but as abnormality of structure, which was observed in 17 panoramic radiographs. Eight panoramic radiographs had two IPFs, including dentigerous cyst, periapical inflammatory lesion, marginal bone loss, cyst in the alveolar bone, thickening of the mucosal lining of sinuses, and idiopathic osteosclerosis.

Figure 1 shows the panoramic radiograph of an 11-year-old female with a cystic lesion involving an unerupted 43 and 44. The lesion has a radiolucent zone of more than 3 mm with well-defined borders around the crowns of tooth 43 and 44.

The age range of the sample was between 7 and 57 years. The median age was 17.2 years and the IQR was 14.7–20.1. Most panoramic radiographs were for those in the 7 to 20 years age category (n = 79). Age did not follow a normal distribution (Shapiro–Wilk test *p* < 0.05). The frequency of the different types of IPFs and age categories are shown in Figure 2. The majority of pathologic findings were most frequent in the 7 to 20 years age group. The results showed no statistically significant differences between the age categories using the Mann–Whitney U test (*p* < 0.832).

A total of 49.2% of the IPFs were detected in the maxilla and 50.8% were seen in the mandible (Figure 3). A statistically significant difference was found (chi-square test *p* < 0.0475).

Other abnormalities detected, in addition to the IPFs, were found in 76 (76%) of panoramic radiographs. These findings were seen in 33 panoramic radiographs with IPFs and 43 that did not have IPFs. Table 3 shows that a total of 134 other findings were detected from the 76 panoramic radiographs, n = 57 (42.5%) findings were in males and n = 77 (57.5%) in females. Impacted teeth were the most common type of other findings (n = 49) and more predominant in females, followed by missing teeth (n = 23), unerupted potential impactions (n = 19), and widening of the periodontal ligament (n = 16). The other findings ranged between one and eight.

## 4. Discussion

The prevalence of IPFs in our study was 38%. Our study findings differed from the study by Bondemark et al., which reports a prevalence of 8.7% [13]. The differences noted may be attributable to ethnic and geographic factors in the sample. Our study findings showed that, of the 38 panoramic radiographs reviewed, 30 had one finding, seven had two findings, and one had three findings. Similarly, Bondemark et al. [13] reported that, of the radiographs analyzed, 27 had one finding, 7 had two findings, and 3 had three findings. A higher prevalence of IPFs compared to our study findings was reported by Ezoddini et al., who found a prevalence of 40.8% in their study [17].

Our study findings showed that every second panoramic radiograph had IPFs. Similar findings were reported by Jadu and Jan, who also note in their study that every second panoramic radiograph had incidental findings [18]. In contrast, every tenth panoramic radiograph had been reported to have IPFs [13], and furthermore, every sixth panoramic radiograph analyzed was also reported to have IPFs [19].

Altered tooth morphology, idiopathic osteosclerosis, and periapical inflammatory lesions were the most common findings in our study. Similarly, idiopathic osteosclerosis have been found to constitute the greatest percentage of findings [13]. The prevalence of altered tooth morphology in our study was 17%, compared to lower prevalence of 0.2% [13] and 1.86% [2] reported. Contrary to our study findings, Goncalves Filho et al. reported a higher prevalence of 71.07% for altered tooth morphology [20]. However, it is a misnomer to regard altered tooth morphology as an IPF in the absence of periapical or periodontal pathoses, but it should be called abnormality of tooth structure.

The prevalence of dentigerous cysts in our study was 1%, compared to the lower prevalence of 0.3%, 0.6%, and 0.76% of dentigerous cysts that has been reported [8,13,21]. In contrast, Hernandez et al. reported a prevalence of 5.34% for dentigerous cyst [22], which was higher than the findings of our study. The prevalence of periapical inflammatory lesions was 8% in our study. Lower prevalence of 0.4% [21,23], 0.5% [24], 1.86% [2], 2% [13], and 2.9% [25] for periapical inflammatory lesions have been found. Other studies have reported higher prevalence of periapical inflammatory lesions ranging from 25–28.6% [18,19,26].

Marginal bone loss had a prevalence of 3% in our study. A lower prevalence of marginal bone loss of 0.2% was reported in the study by Bondemark et al. [13]. Other studies have also reported lower prevalence of 0.4% [24], 0.233% [2], and 1.85% [21] compared to the findings of our study.

Cysts in the alveolar bone had a prevalence of 2% in our study. Bondemark et al. report a much lower prevalence of 0.4% for cysts in the alveolar bone [13]. Granlund et al. also report a slightly lower prevalence of 1.01% for cysts in the alveolar bone [2] compared to the findings in our study.

Our study found that the prevalence of idiopathic osteosclerosis to be 13%. Cederhag et al. report a prevalence of 20% for idiopathic osteosclerosis [26], higher that our study findings. A lower prevalence of 4.4% for idiopathic osteosclerosis was reported in the study of Bondemark et al. [13]. Other studies have reported prevalence ranging from 2.7% to 10.7% [2,18,19,20,23,24]. Higher prevalence of idiopathic osteosclerosis has been reported to be caused by an increase in water fluoridation, which results in osteosclerotic lesions forming [22].

The prevalence of the thickening of mucosal lining of sinuses was 3% in our study. Bondemark et al. also report a prevalence of 3% for thickening of the mucosal lining of sinuses [13], whereas Vaseemuddin reports a prevalence of 1.5% [27]. Contrary to our findings, other studies have reported higher prevalence that range from 7% to 21.22% [18,25,26,28]. The differences in the prevalence of thickening of the mucosal lining of sinuses may be due to differences in geographical locations, seasonal changes, and study populations [29].

The results of our study showed a predominance of females (59%) compared to males (49%). Similarly, Cral et al. report 56.4% females and 43.6% males in their study sample [21]. However, our findings showed that males had a higher prevalence of IPFs (55.3%) compared with females (44.7%). Similar findings of males having a higher prevalence of IPFs than females have been reported [17]. Yet, other studies have found IPFs to be higher in females [11,13,30]. The frequency of the thickening of mucosal lining of maxillary sinuses was almost similar in males (2%) and females (1%). Similar findings have been reported regarding maxillary sinus pathology in females (1.3%) and males (2.32%) [31]. Our study found idiopathic osteosclerosis to be more common in males compared to a previous study that reported predominance in females [29]. The relationship of IPFs with gender was not statistically significant (*p* > 0.107).

The median age of our sample was 17.2 years. The majority of panoramic radiographs were for those in the 7 to 20 years age category (n = 79). This age range is expected since our study sample was selected from the pretreatment orthodontic panoramic radiographs. This is the age by which patients seek and require orthodontic treatment; therefore, examining their radiographs thoroughly beyond just malocclusion is important for comprehensive diagnosis and treatment planning. All IPFs were more frequent in the 7 to 20 years age category. Young patients (7 to 20 years age category) had a higher prevalence (n = 27; 27%) of IPFs in our study compared to older patients (21 to 40 years and 41 to 57 years age categories), who had lower prevalence (n = 10; 10%) and (n = 1; 1%), respectively. Similar findings in panoramic radiographs were also reported to be more common in young patients [22]. However, older patients (20 to 30 years) have been found to a have a higher prevalence of pathologic findings at 62.4% compared to those found in the 12 to 20 years age category at 37.6% [30]. The variations in patient age categories and variations in diagnostic criteria and definition may also partly explain the differences in the results. The relationship between IPFs with age was not statistically significant (*p* > 0.832). Jadu and Jan also found no significant relationship in incidental findings with age [18]. In a different study, a significant association (*p* < 0.001) was found between age and the presence of pathology [22].

The majority of the IPFs (50.8%) were located in the mandible compared to the maxilla (49.2%), and similar results of mandible predominance have been reported [19]. Contrary results have been reported where the mandible had the lower prevalence of findings compared to the maxilla [30]. The results of the current study further show that the mandible had a higher frequency of periapical inflammatory lesions than the maxilla. This can be attributed to the maxilla having a greater chance of superimposition of anatomical structures compared to the mandible. Furthermore, idiopathic osteosclerosis was found in the posterior region of the mandible. Similarly, previous studies reported that almost all of the idiopathic osteosclerosis was seen in the mandible [23,24].

Some clinically visible abnormalities expected to be generally noted on panoramic radiograph assessment, such as crowding, supernumerary teeth, impacted teeth, and spacing were found in 76% of the sample. Impacted teeth (n = 49) and missing teeth (n = 23) were the most frequent findings. Our study findings reported a 49% prevalence for impacted teeth, and more common in females, compared to the study by MacDonald and Yu, which reports impacted teeth to be more common in males [23]. The prevalence of impacted teeth has been reported to range from 4.4% to 29.6% [11,20,23,32].

The study had to cross the hurdle of missing records. This limitation was overcome by omitting those records with missing data and adding records until the sample size was achieved. This approach is known as available case analysis [33]. The sample size gender was not equally distributed; therefore, the results cannot be generalized that some IPFs are more frequent in one of these groups. The study was undertaken using pretreatment orthodontic panoramic radiographs. No examination was made of the post-treatment radiographs to evaluate the management of the findings. The incidental findings of dentigerous cysts, periapical inflammatory lesions, and cysts in the alveolar bone were based on only the radiographic findings in this study and are, therefore, burdened with the uncertainty that comes with radiographic assessments made without histologic confirmation of these lesions. It was also difficult to determine whether the missing teeth recorded in our study were congenital missing teeth or teeth that had been extracted, since this was a radiographic review and information on the missing teeth had not been obtained.

The ability to detect incidental findings in orthodontic pretreatment panoramic radiographs should be a skill that each clinician has in practice. Continued education and training of clinicians is recommended in examining and reporting panoramic radiographs thoroughly as they are commonly used in clinical dentistry. Detection of IPFs that may need odontological or medical intervention should be followed up for comprehensive management of patients. Further studies should be conducted using a study sample representative of all panoramic radiographs of the dental school for further audit and quality assurance.

## 5. Conclusions

Panoramic radiograph is a preferred routine diagnostic tool for evaluation of the dentition in contemporary dental practice. The additional advantage is the detection of incidental findings, including pathology, which may remain undetected if other radiographic views are taken, thus proving useful in detecting hidden findings. The detection of IPFs in our study shows the importance for clinicians to examine panoramic radiographs thoroughly for IPFs beyond the orthodontic counting of teeth. The presence of IPFs, such as periapical inflammatory lesions, dentigerous cysts, marginal bone loss, and cysts in the alveolar bone, can have an impact on diagnosis and treatment planning, especially in orthodontics.

## Figures and Tables

**Figure 1 ijerph-20-03479-f001:**
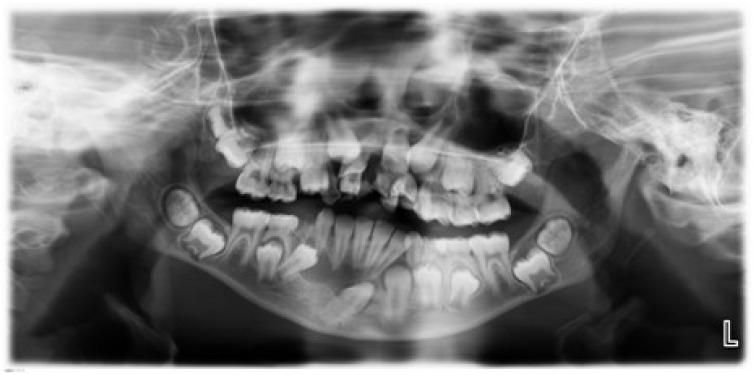
Panoramic Radiograph of an 11-year-old female with a dentigerous cyst.

**Figure 2 ijerph-20-03479-f002:**
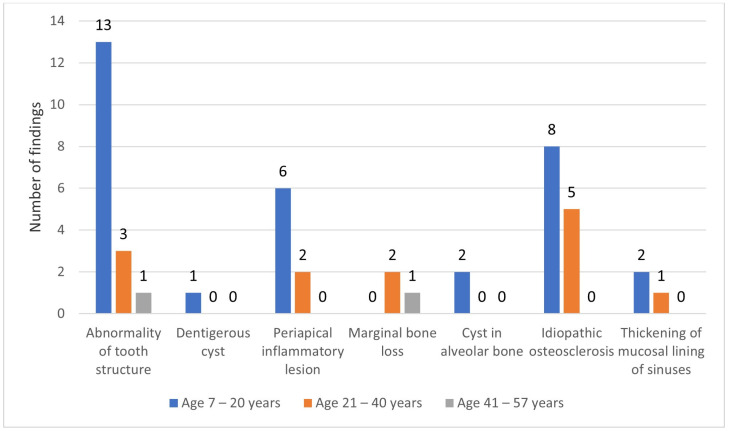
Frequency of incidental pathologic findings by age category.

**Figure 3 ijerph-20-03479-f003:**
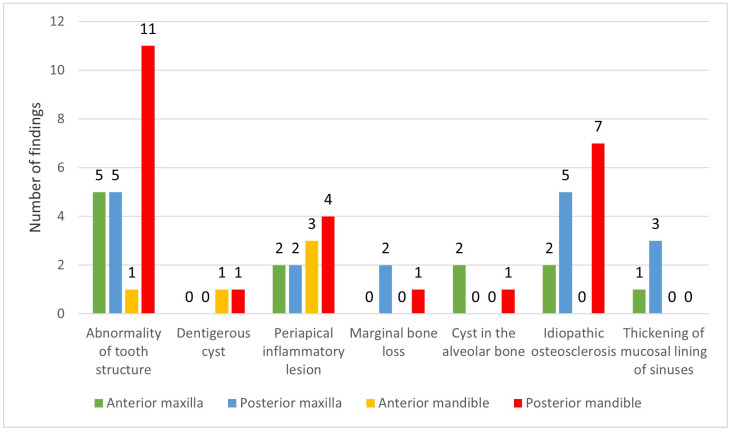
Incidental pathologic findings according to the jaw location.

**Table 1 ijerph-20-03479-t001:** Radiographic Criteria used for Recording of IPFs.

Location	Finding	Criteria
Tooth/tooth associated	Altered morphology	Radiographically unusual shape or size of the tooth
	Dentigerous cyst	Radiolucent zone of more than 3 mm with well-defined borders around the crown of an unerupted tooth, epicenter just above the crown, and the cyst attaches at the cemento-enamel junction
	Periapical inflammatory lesion	Radiolucent or radiopaque change in association with a tooth apex with radiographically interrupted lamina dura
	Marginal bone loss	Distance between the cementoenamel junction and the alveolar bone crest larger than 2 mm
Alveolar bone	Cyst	Centrally located radiolucency within the bone, round or oval and with well-defined borders and corticated thin radiopaque line
	Idiopathic osteosclerosis	Radiopaque change, i.e., dense trabeculae calcifications with well- or ill-defined borders in the surrounding bone
Sinuses	Thickening of mucosal lining	Density along the sinus floor or generalized density of the maxillary sinus or cystic (oval, well defined) density in any area of the sinus

Source: Bondemark et al. [13].

**Table 2 ijerph-20-03479-t002:** Incidental pathologic findings and abnormality of structure in males and females.

	Findings	Male	Female	Total
Abnormality of tooth structure	Altered tooth morphology	8 (17%)	9 (19.1%)	17 (36.1%)
IPF	Dentigerous cyst	0 (-)	1 (2.1%)	1 (2.1%)
	Periapical inflammatory lesion	5 (10.6%)	3 (6.4%)	8 (17%)
	Marginal bone loss	3 (6.4%)	0 (-)	3 (6.4%)
	Cyst in the alveolar bone	0 (-)	2 (4.3%)	2 (4.3%)
	Idiopathic osteosclerosis	8 (17%)	5 (10.7%)	13 (27.7%)
	Thickening of mucosal lining of sinuses	2 (4.3%)	1 (2.1%)	3 (6.4%)
	Total	26 (55.3%)	21 (44.7)	47 (100%)

**Table 3 ijerph-20-03479-t003:** Other abnormalities in males and females.

Other Findings	Male	Females	Total
Impacted teeth	18 (13.43%)	31 (23.13%)	49 (36.6%)
Widening of periodontal ligament	6 (4.47%)	10 (7.46%)	16 (11.9%)
Pulp stones	3 (2.2%)	5 (3.7%)	8 (5.9%)
Rotated teeth	3 (2.2%)	2 (1.5%)	5 (3.7%)
Missing teeth	11 (8.2%)	12 (9%)	23 (17.2%)
Unerupted potential impaction	10 (7.5%)	9 (6.7%)	19 (14.2%)
Crowding	2 (1.5%)	6 (4.5%)	8 (6%)
Spacing	3 (2.2%)	1 (0.7%)	4 (3%)
Supernumerary teeth	1 (0.7%)	0 (-)	1 (0.7%)
Retained deciduous teeth	0 (-)	1 (0.7%)	1 (0.7%)
Total	57 (42.5%)	77 (57.5%)	134 (100%)

## Data Availability

Data available from the University of the Witwatersrand website, http://wiredspace.wits.ac.za.

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
