# Peer review of "Incidental Pathologic Findings from Orthodontic Pretreatment Panoramic Radiographs"

_ijerph, 2023, doi:10.3390/ijerph20043479_

Round 1
Reviewer 1 Report
It is an interesting paper, I like it; however, some sections could be improved as follow:
In results section, the percentages should be added in figures 1 and 2 (frequency and percentage) at the top of each column for a more appropriate understanding of what is mentioned in the discussion section. They could also be replaced by tables (another option).
In the discussion section, before discussing each IPFs, the results are mentioned with a percentage, written as a result. The comparisons are fine but the results in this section should refer to higher or lower than what is reported by other authors.
It is important to discuss that there is no equality in the distribution of the sample by gender or by age, therefore it cannot be affirmed that some IPFs are more frequent in one of these groups, and also include in limitations of the study pragraph.
Additional results were found for other pathologies that do not include the IPFs classification used, perhaps they should be reported as additional results in the results section and discuss something about it in the corresponding section.
There are too many conclusions, some of them are results and others do not correspond to the objective of the study. The most important are the last 3, although the last one should be redirected to the conclusion and not described as a result,
Author Response
See attached Reviewer's comments and Responses.

Reviewer 2 Report
Thank you for allowing me to review this scientific article. Here are some of the corrections needed:
1. Abstract: the conclusions are too general, they are discussed according to the results
2. The introduction is too long, I think it should be revised
3. Methods: where were the patients selected from? The inclusion-exclusion criteria are not clear.
4. It should be explained more clearly: how many radiographs were taken, how many had IPF, the distribution according to sex, eg.
5. Table 1 should be moved immediately after the paragraph (line 123).
6. 100 patients were evaluated, what does the difference from 47 to 100 represent?
7. In figure 1, the explanations are not clear enough and are quite difficult to follow
8. The explanation in table 2 is unclear
9. The order should be changed: first the demographic characteristics, then the frequency of lesions, and finally the radiological image.
10. Discussions are too long
11. Conclusions: do not discuss the results again.
Author Response
See attached Reviewer's comments and responses

Reviewer 3 Report
This is an interesting study, however there are some concerns to be addressed:
The inclusion and exclusion criteria should be further detailed.
What is the novelty of the paper?
What is the clinical relevance?
Figure 2 can be embedded in the manuscript
The conclusion is a repetition of the results, it should be compressed and related to the relevance of the study, focusing on clinical data.
References are too old, they should be after 2015.
Author Response

(The authors gave the same response as above.)

Reviewer 4 Report
1. Since incidental pathological findings (IPF) in panoramic radiographs (OPG) can range from simple marginal bone loss and PDL pace widening, till odontogenic tumors and cysts, it would be clear to the reader if the IPFs which were detected as part of the study were defined/delineated in the introduction and study objectives.
2. Similar to the previous comment, the Table 1 which gives the radiographic criteria for diagnosis of IPFs should be aligned with the pathologies defined in the introduction.
3. Add a short note in the methods section about how the intra- and inter-observer reliability and agreement were arrived at. What was the sample size used to arrive at the above parameters?
4. How were the radiographs included in the study to reach the sample size? Was it through random sampling or systematic sampling or sequential sampling?
5. Also, it has been mentioned in the discussion that more than 75% of the study sample were below the age of 20 years. How could that mean a representative sample of identifying IPFs in the general population?
6. Since it was a retrospective study, how were the acquired radiographs standardized in terms of the parameters of exposure.
7. Based on Table 2, “Altered tooth morphology” is the most prevalent IPF. But is altered tooth morphology without any periapical/periodontal pathoses really a pathological finding?
Author Response
see attached Reviewer's comments and responses

Reviewer 5 Report
The authors submitted a manuscript entitled “Incidental pathologic findings from orthodontic pretreatment panoramic radiographs”. The manuscript has the potential for publication, but it needs some revisions.
Please see my recommendations below:
Introduction
· The Introduction section is too long and contains a lot of information that belongs to the Discussion section. I recommend emphasizing and describing the various pathologic findings that can be identified on panoramic radiographs.
· The paragraphs between Lines 57 and 89 should be moved to the Discussion section.
Material and Methods
· Is well structured and detailed.
Results
· Figure 1 is not representative of this study. I this that this Figure must be erased and replaced with a better panoramic radiograph.
· Issues with this figure and its interpretation:
o Point A should describe the periapical inflammatory lesions identified in teeth 1.2, 2.1, 2.2 but the arrow connected to point A indicates teeth 1.3 and 1.5.
o There is no periapical inflammatory lesion in teeth 1.2, 2.1, 2.2. The periodontal space can be traced with no problems. These teeth have no caries and no fillings. The identified IPF does not exist.
o Point B should indicate the altered tooth morphology of teeth 3.3 and 3.4, yet the arrow points at tooth 4.2.
o In addition, tooth 3.5 has a normal morphology, identical to tooth 4.5.
· The authors should be more careful when submitting a manuscript for a journal with an IF of 4.615.
· The authors should examine in detail every IPF before diagnosing it.
Discussion
· The paragraphs between Lines 57 and 89 should be added here.
· The limitations of the study and the recommendations should not be separate subchapters and should be added as final paragraphs of the Discussion section to respect the journal’s authors' instructions.
Conclusions
· This section should be shortened and reduced to 1 or 2 paragraphs.
· This section should not contain any numerical results.
Best regards!
Author Response

(The authors gave the same response as above.)

Round 2
Author Response
Thank you for the comments, see attached responses.

Reviewer 4 Report
Although the others have claimed in their response to have addressed to all my review comments, I see no revisions done to the manuscript in terms of my queries related to:
1. Methodology behind inter-/intra-observer reliability estimation - I had asked about how it was done and there is no explanation mentioned about that in the revised manuscript. Authors have to describe the pilot sample size and parameters used while determining inter-/intra-observer reliability, and not simply claim a kappa statistic.
2. Is "altered tooth morphology" an incidental pathological finding? - with regard to this comment of mine, the authors themselves have responded that it is a misnomer and have discussed about in the discussion section. I see nothing of that sort in the revised manuscript. Moreover, if the authors agree that "altered tooth morphology", should have actually been mentioned as "abnormality in tooth structure", why not incorporate the same terminology throughout the manuscript?
Author Response
Thank you for the comments. See attached responses

Reviewer 5 Report
Dear authors,
Thank you for the revisions made.
Still, Figure 1 (panoramic radiograph) was not changed.
I strongly recommend replacing it with an adequately interpreted panoramic radiograph, before accepting this article.
Best regards!
Author Response

(The authors gave the same response as above.)
